

# Sources and processes of iron aerosols in a megacity of Eastern China

Yanhong Zhu[1], Weijun Li[1*], Yue Wang[1], Jian Zhang[1], Lei Liu[1], Liang Xu[1], Jingsha Xu[2], Jinhui Shi[3], Longyi Shao[4], Pingqing Fu[5], Daizhou Zhang[6], Zongbo Shi[7*]

[1]Department of Atmospheric Sciences, School of Earth Sciences, Zhejiang University, Hangzhou 310027, Zhejiang, China
[2]Department of Chemistry, University of Warwick, Coventry, CV4 7AL, UK
[3]Key Laboratory of Marine Environmental Science and Ecology, Ocean University of China, Ministry of Education of China, Qingdao 266010, China
[4]State Key Laboratory of Coal Resources and Safe Mining, China University of Mining and Technology, Beijing 100086, China
[5]Institute of Surface-Earth System Science, School of Earth System Science, Tianjin University, Tianjin 300072, China
[6]Faculty of Environmental and Symbiotic Sciences, Prefectural University of Kumamoto, Kumamoto 862-8502, Japan
[7]School of Geography, Earth and Environmental Sciences, University of Birmingham, Birmingham B15 2TT, UK

*Correspondence to:* Weijun Li (liweijun@zju.edu.cn), Zongbo Shi (Z.Shi@bham.ac.uk)

**Abstract.** Iron (Fe) in aerosol particles is a major external source of micronutrients for marine ecosystems, and poses a potential threat to human health. To understand these impacts of aerosol Fe, it is essential to quantify the sources of dissolved and total Fe. In this study, we applied a receptor modelling for the first time to apportion the sources of dissolved and total Fe in fine particles collected under five different weather conditions in Hangzhou megacity of Eastern China, which is upwind of East Asian outflow. Results showed that Fe solubility (dissolved to total Fe) was the largest in fog days (6.3 ± 2.6%), followed by haze (4.6 ± 1.9%), dust (2.1 ± 0.7%), clear (1.7 ± 0.6%), and rain (0.8 ± 0.3%) days. Positive Matrix Factorisation (PMF) analysis suggested that industrial and traffic emissions were the two dominant primary sources of dissolved and total Fe during haze and fog days, but dust was the dominant source in dust days. About 15% of dissolved Fe was associated with secondary sources during haze and fog days, although it was less than 5% during dust and clear days. Transmission electron microscopy analysis of individual particles showed that approximately 76% and 87% of Fe-containing particles were internally mixed with acidic sulfates and nitrates in haze and fog days, respectively. Our results indicated that aqueous surface of aerosol particles promotes heterogeneous reactions between acidic species and Fe aerosol, contributing to higher Fe solubility during fog and haze days.

## 1  Introduction

The deposition of atmospheric aerosols is a major external source of iron (Fe) in the ocean (Li et al., 2017; Pinedo-González et al., 2020; Yang et al., 2020). Fe is an essential micronutrient that can impact phytoplankton primary productivity, thereby modulating marine ecosystems, global carbon cycling, and climate (Jickells et al., 2005; Tagliabue et al., 2017; Matsui et al.,



2018; Lei et al., 2018). In addition, atmospheric Fe-containing particles have an adverse effect on human health by generating reactive oxygen species (ROS) (Abbaspour et al., 2014), and can convert S(IV) to S(VI) by catalytic oxidation for

atmospheric sulfate ($SO_4^{2-}$) production (Alexander et al., 2009). These roles of Fe largely depend on the atmospheric Fe solubility (Shi et al., 2012; Baker et al., 2021). Unfortunately, field observations on atmospheric Fe solubility are still limited, and the available data show a wide range of Fe solubility (0.02 % to 98 %) in different atmospheric environments (Schroth et al., 2009; Shi et al., 2012; Oakes et al., 2012; Myriokefalitakis et al., 2015).

There are two major processes that can significantly increase Fe solubility in atmospheric aerosols, including aerosol primary emissions and atmospheric acidification processes (Shi et al., 2012). Dissolved Fe can be derived from natural and anthropogenic sources, such as mineral dust, fossil fuel combustion, biomass burning, and traffic exhaust (Chen et al., 2012; Pant et al., 2015; Conway et al., 2019; Rathod et al., 2020; Ito et al., 2020). Although natural emissions having a high emission flux, their contribution to Fe solubility is less than 1% (Schroth et al., 2009). Recent studies have highlighted

anthropogenic sources due to their high contribution to Fe solubility. For example, Schroth et al. (2009) suggested that Fe solubility was less than 1% of the iron in arid soils, while oil combustion emissions had a pronounced effect on Fe solubility (77–81%); Oakes et al. (2012) studied Fe solubility in anthropogenic source emission samples and found that Fe solubility was 0.06% in coal fly, 46% in biomass burning, 51% in diesel exhaust, and 75% in gasoline exhaust. These results imply that an increase in relative amounts of aerosols from anthropogenic sources could lead to the increase in Fe solubility.


There are a number of atmospheric processes, which can affect Fe solubility in atmospheric aerosol particles. However, the most important process is the mobilization of Fe in acidic solution on the surface of aerosol particles, because acidic pH can trigger faster Fe dissolution and increase Fe solubility (Shi et al., 2011; Maters et al., 2017; Li et al., 2017; Zhou et al., 2020). Particulate composition and the ambient relative humidity (RH) can determine phase state of aerosol particles (Li et al.,

2021a). When ambient RH is above 50%, the surfaces of secondary aerosol particles start to change from solid to wet or liquid state (Sun et al., 2018), which can take up precursor acid gases and form acidic aqueous medium on the surface of aerosol particles (Li et al., 2016; Zhang et al., 2021). In a word, acid gases such as $SO_2$ and $NO_2$ from anthropogenic sources can convert into acidic aerosols on aerosol particles and further promote the conversion of Fe from undissolved to dissolved form, thereby increasing Fe solubility (Li et al., 2017; Zhang et al., 2019a; Yang et al., 2020; Wong et al., 2020).


The two major contributors mentioned above (aerosol primary sources and atmospheric acidification processes) to Fe solubility are strongly associated with weather condition. Because weather condition can change ventilation efficiency (such as wind, frontal passages, and boundary layer height), relative humidity, and loss ratios of chemical conversion (Zhang et al., 2018). Recently, Shi et al. (2020) found that different levels of Fe solubility are closely related with different weather

conditions in one coastal city. However, to our knowledge, studies that have attempted to investigate Fe solubility under different weather conditions in the megacity are still sparse in the world, even though the sources of aerosol Fe (such as coal



combustion, vehicle emission, industry emission) are densely distributed in megacities (Zhang et al., 2019b). Therefore, to better understand how aerosol primary sources and atmospheric acidification processing influence Fe solubility in the megacity, the planned studies should be conducted under different weather conditions.


In this study, we collected atmospheric fine particles (PM$_{2.5}$) and individual particle samples in haze, fog, dust, clear, and rain days at Hangzhou, a megacity of Yangtze River Delta (YRD), which is one of the largest modern megacity-clusters in China. This study characterized Fe content and solubility under haze, fog, dust, clear, and rain weather conditions, and discussed the impacts of primary sources and atmospheric acidification processes on Fe solubility.

## 2    Methodology

### 2.1    Sampling Site

The sampling site was located in the Zijingang Campus of Zhejiang University in Hangzhou (120°12′ E, 30°16′ N), a megacity in the YRD, China (Fig. S1). Industrial emissions are relatively low in Hangzhou comparison to other megacities in China, but traffic emissions are serious (Xu et al., 2020). In addition, pollutants emitted in surrounding regions and northern China can be transported to Hangzhou city (Liu et al., 2021).

### 2.2    Sample collection

PM$_{2.5}$ aerosol and individual particle samples were collected under haze, fog, dust, clear, and rain weather conditions between November 2018 and January 2020. Details on the sampling periods were shown in Table S1. The definitions of haze, fog, dust, clear, and rain weather conditions were shown in Table S2. When the duration of haze, fog or dust exceeded 70% of the collection time of a sample, the sample was classified as a haze, fog, or dust sample. Totally, there were 30 haze samples, 28 fog samples, 12 dust samples, 30 clear samples, and 9 rain samples in this study.

One TH-16A Intelligent sampler (Wuhan Tianhong Corporation, China) with a flow rate of 100 L min$^{-1}$ was used to collect PM$_{2.5}$ samples on 90 mm diameter quartz filters for 11.5 h (daytime: 08:30-20:00; nighttime: 20:30-08:00 (next day)). The sampler was installed on the rooftop of a four-story teaching building (approximately 20 m above the ground) on the Zijingang campus of Zhejiang University. All quartz filters were firstly baked at 600 ℃ in a muffle furnace for 4 h to remove contaminants. Then, these filters were conditioned in a room with temperature of 20 ± 1 ℃ and RH of 50 ± 2%. After 24 h, these filters were weighed using a Sartorius analytical balance (detection limit 0.001 mg). After sample collection, the loaded filters were similarly conditioned and weighed in order to determine PM$_{2.5}$ mass concentrations. Daytime and nighttime blank samples were collected by the same method with real samples, but without operating the sampler. The collected filters were preserved in a freezer at -4 ℃ until further analysis.



Individual particle samples were collected four times at 8:00, 12:00, 18:00 and 0:00 in each day. The sampler is a single-stage cascade impactor with a 0.5 mm diameter jet nozzle and a flow of 1.0 L min$^{-1}$. The samples were collected on copper grids coated with carbon film. According to weather and visibility, the sampling duration spanned 30 s to 8 min. The collection efficiency is 50% for particles with an aerodynamic diameter of 0.1 μm and a density of 2 g cm$^{-3}$. After sampling, the grids were placed in a dry plastic tube and stored in a desiccator at 25 ℃ and 20 ± 3% RH to minimize exposure to ambient air.

## 2.3 Elemental analysis

Element concentrations were determined by Energy Dispersive X-Ray Fluorescence (EDXRF) spectrometer (Epsilon 4, PANalytical). In this method, element concentrations on a given elemental map were measured. The measured values firstly divided by the elemental map area, then multiplied by the total sample area to obtain element concentrations of the sample. Because quartz filter contains a large amount of silicon (Si), Si measured by EDXRF is not used in this study. Elements including Na, Mg, Al, P, S, Cl, K, Ca, Ti, V, Cr, Mn, Fe, Co, Ni, Cu, Zn, Ga, As, Se, Sr, Ba, and Pb were measured. The National Institute of Standards and Technology (NIST) standard was used as reference material for standardizing the instrument. The analysis values of NIST standard were given in Table S3, showing that the relative errors between measured and standard value for the standard samples were less than 10%. The average element concentrations of field blank samples (n = 4) were well below those of the samples (Table S3), indicating that no significant contribution of blank subtraction to the observed concentrations. The elemental concentrations used in this study were corrected by subtracting the filter blank values.

## 2.4 Sample preparation and analysis of dissolved Fe

Chemical analysis of dissolved Fe was conducted using the ferrozine technique described by Viollier et al. (2000). Sample extraction and analysis were on the basis of Majestic et al. (2006) and Oakes et al. (2012). We conducted analysis as follows: (1) Half of the sample filters were placed in clean tubes with 20 mL ammonium acetate (0.5 mM, pH = 4.3). Then, the tubes were placed in an ultrasonic bath for 60 min. The extractions were filtered through a 0.22 μm PTFE syringe filter to remove undissolved particles. (2) The concentrated HCl was immediately added to the filtrate to adjust pH equal to about 1, and then the filtrate was store in the refrigerator. (3) Before the storage solution start to analyze, a solution of 0.01 M ascorbic acid was added to the filtrate to reduce Fe(III) to Fe(II), and held for 30 minutes to ensure complete Fe reduction. (4) 0.01 M ferrozine solution was added to the filtrate. (5) Ammonium acetate buffer (pH = 9.5) was added to the filtrate, making pH between 4 and 9. Light absorption of the mixture was immediately measured by Ultraviolet-visible Spectrophotometer at 562 nm (max light absorption of Fe(II)-Ferrozine complex) and 700 nm (background measurement) to yield dissolved Fe measurement. SigmaUltra-grade ammonium Fe(II) sulfate was used for Fe(II) standards. The concentration of Fe(II) obtained from the standard curve was the concentration of dissolved Fe. The detection limit of the method for Fe(II) was 0.11 ng m$^{-3}$, calculated as three times the standard deviation of filter blank values (n = 9). The concentrations of Fe(II) in the



field blanks were all below the detection limit, and the data reported in this study were corrected by subtracting the filter blank values.

## 2.5 Individual particle analysis

Individual particle samples were analyzed with a JEOL JEM-2100 transmission electron microscope (TEM) operated at 200 kV. Elemental composition was semi-quantitatively determined by an energy-dispersive X-ray spectrometer (EDS) that can

detect elements heavier than carbon (C). Copper (Cu) was excluded from the analyses, because the TEM grids are made of Cu. The relative percentages of the elements were estimated based on the EDS spectra acquired through the INCA software (Oxford Instruments, Oxfordshire, UK). The distribution of aerosol particles on TEM grids was not homogeneous: coarser particles occur near the center and finer particles are on the periphery. Therefore, to be more representative, four areas were chosen from the center to periphery of the sampling spot on each grid. The projected areas of individual particles were

determined using iTEM software (Olympus Soft Imaging Solutions GmbH, Germany), the standard image analysis platform for electron microscopy.

## 2.6 Water-soluble inorganic ions, organic carbon, and elemental carbon, SO$_2$, and NO$_2$

The concentrations of water-soluble inorganic ions including Na$^+$, NH$_4^+$, K$^+$, Mg$^{2+}$, Ca$^{2+}$, F$^-$, Cl$^-$, NO$_3^-$, and SO$_4^{2-}$ were obtained by an ion chromatograph (Dionex ICS 600, ThermoFisher Scientific). Detailed descriptions about filter extraction

and analysis were given in Zhu et al. (2015). Organic carbon (OC) and elemental carbon (EC) were analyzed by a Sunset Laboratory carbon analyzer with the thermal-optical transmittance method. The concentrations of SO$_2$ and NO$_2$ were obtained from the website of https://www.aqistudy.cn/.

## 2.7 Enrichment factor (EF)

The EF is an important index for quantitative evaluation of the enrichment degree of elements. This factor is used to

determine the elements in the aerosol are derived from natural or anthropogenic sources. EF is defined (Rudnick and Gao, 2003) as follow:

$$EF_i = \frac{(C_i / C_{ref})_{aerosol}}{(C_i / C_{ref})_{crust}}$$

Here, $C_i$ is the concentration of element i in the aerosol or crust; $C_{ref}$ is the concentration of reference element in the aerosol or crust. Al is chose as reference element. EF value < 10 suggests element has crustal sources, whereas EF value > 10 is

related to anthropogenic sources.





## 2.8 Positive matrix factorisation (PMF)

The USEPA PMF 5.0 model was used to identify sources of dissolved Fe. A detailed description about PMF 5.0 was given in the user manual (USEPA, 2014). Two input files are required to initiate the model: one containing concentration values and one containing uncertainty values for each species. Uncertainty was determined as follows (Polissar et al., 1998):

$$\text{If } C_i \leq MDL, \text{ Unc } = \frac{5}{6} \times MDL;$$

$$\text{If } C_i > MDL, \text{ Unc } = \sqrt{(\text{Error Fraction} \times \text{concentration})^2 + (0.5 \times MDL)^2};$$

where $C_i$ is the concentration value; MDL is the method detection limit; Unc is the uncertainty. The principals of PMF running and species choice have been described in the PMF 5.0 User Manual and our previous study (Zhu et al., 2017). In this study, 100 samples were used to run PMF model. $PM_{2.5}$, OC, EC, $SO_4^{2-}$, $NO_3^-$, $NH_4^+$, Mg, Al, K, Ca, Ti, Cr, Mn, Co, Ni, Cu, Zn, As, Se, Sr, Ba, Pb, dissolved Fe, and undissolved Fe (= total Fe - dissolved Fe) were used for PMF analysis and six
factors were resolved as the optimal solution, the selection of which is described in Supplemental Information in details. Dissolved Fe was set as total variable, and $PM_{2.5}$ was set as weak variable. The changes in Q values can provide insight into the rotation of factors. The $Q_{Robust}$ (2392.94) was close to $Q_{True}$ (2474.51), suggesting PMF results can reasonably explain potential sources of dissolved Fe. Since the number of samples should be 3 times higher than the number of species used in PMF, accurate PMF results could be obtained, so we used the sum of all samples in haze, fog, dust, and clear weather
conditions to run PMF model.

## 3 Results and Discussion

### 3.1 Pollution levels

Figure 1a shows the variations of $PM_{2.5}$ concentrations under haze, fog, dust, clear, and rain weather conditions during the sampling periods. The average $PM_{2.5}$ concentration was the highest at $98.3 \pm 20.6$ µg m$^{-3}$ in haze days, followed by $59.3 \pm$
$11.1$ µg m$^{-3}$ in dust days, $57.5 \pm 26.9$ µg m$^{-3}$ in fog days, $33.6 \pm 14.5$ µg m$^{-3}$ in clear days, and $31.4 \pm 8.1$ µg m$^{-3}$ in rain days. About 100%, 29%, and 8% of $PM_{2.5}$ concentrations in haze, fog, and dust days were higher than the Grade II national $PM_{2.5}$ standard of 75 µg m$^{-3}$ (24 h average standard, GB 3095-2012, China), respectively. However, all of $PM_{2.5}$ concentrations in clear and rain days were lower than the $PM_{2.5}$ Grade II standard. $PM_{2.5}$ concentrations differed significantly according to weather conditions (p < 0.01, independent sample T test, Table S4).


Figures 1b and 1c show concentration variations of gaseous pollutants (e.g., $SO_2$ and $NO_2$). $SO_2$ concentration was the highest at $11.0 \pm 2.7$ µg m$^{-3}$ in haze days, followed by $8.0 \pm 1.9$ µg m$^{-3}$ in fog days, $7.8 \pm 1.1$ µg m$^{-3}$ in dust days, $6.6 \pm 1.1$ µg m$^{-3}$ in clear days, and $5.6 \pm 0.8$ µg m$^{-3}$ in rain days. $SO_2$ concentrations under these five kinds of weather conditions were all much lower than the Grade II national $SO_2$ standard of 150 µg m$^{-3}$ (24 h average standard, GB 3095-2012, China). In



addition, $SO_2$ concentrations differed significantly according to weather conditions ($p < 0.01$, independent sample T test, Table S4), except between haze and fog days ($p > 0.05$). $NO_2$ showed the highest concentration at $64.2 \pm 12.2$ µg m$^{-3}$ in haze days, followed by $58.4 \pm 19.0$ µg m$^{-3}$ in fog days, $51.8 \pm 9.9$ µg m$^{-3}$ in dust days, $40.6 \pm 17.1$ µg m$^{-3}$ in clear days, and $34.3 \pm 8.9$ µg m$^{-3}$ in rain days. About 10% and 15% of $NO_2$ concentrations in haze and fog days were higher than the Grade II national $NO_2$ standard of 80 µg m$^{-3}$ (24 h average standard, GB 3095-2012, China), respectively. During dust, clear, and rain days, $NO_2$ concentrations were always below the Grade II national $NO_2$ standard. $NO_2$ concentrations differed significantly according to weather conditions ($p < 0.01$, independent sample T test, Table S4), except between haze and fog days ($p > 0.1$), clear and rain days ($p > 0.4$).

Figure 1d shows the variations of total concentration of all detected inorganic ions. The average concentration of all detected inorganic ions was the highest at $30.7 \pm 14.1$ µg m$^{-3}$ in fog days, followed by $23.8 \pm 12.2$ µg m$^{-3}$ in haze days, $20.0 \pm 5.8$ µg m$^{-3}$ in dust days, $19.4 \pm 5.0$ µg m$^{-3}$ in rain days, and $17.4 \pm 8.5$ µg m$^{-3}$ in clear days. The concentrations of all detected inorganic ions differed significantly according to weather conditions ($p < 0.01$ or $0.05$, independent sample T test, Table S4).

Figure 1e shows the variations of the concentrations of all detected elements. The average concentration of all detected elements was the highest at $14.4 \pm 4.7$ µg m$^{-3}$ in dust days, followed by the similar concentrations in haze ($8.0 \pm 1.6$ µg m$^{-3}$), fog ($8.2 \pm 1.8$ µg m$^{-3}$), and clear ($8.3 \pm 1.0$ µg m$^{-3}$) days, as well as $7.5 \pm 1.5$ µg m$^{-3}$ in rain days. However, the concentrations of all detected elements differed significantly according to weather conditions ($p < 0.01$ or $0.05$, independent sample T test, Table S4).

## 3.2 Fe content and solubility

The average concentrations of total and dissolved Fe were $765.4 \pm 283.4$ and $34.3 \pm 15.3$ ng m$^{-3}$ in haze days, $861.6 \pm 378.3$ and $55.4 \pm 36.1$ ng m$^{-3}$ in fog days, $2945.9 \pm 735.1$ and $57.4 \pm 12.4$ ng m$^{-3}$ in dust days, $647.5 \pm 192.1$ and $11.1 \pm 6.0$ ng m$^{-3}$ in clear days, $652.5 \pm 306.5$ and $5.4 \pm 4.3$ ng m$^{-3}$ in rain days (Figure 2a and 2b). Total Fe concentrations differed significantly according to weather conditions ($p < 0.01$ or $0.05$, independent sample T test, Table S4), except between haze and clear days ($p > 0.1$), and between fog and clear days ($p > 0.5$). Dissolved Fe concentrations differed significantly according to weather conditions ($p < 0.01$ or $0.05$). The contributions of total and dissolved Fe concentrations to $PM_{2.5}$ concentration are shown in Table 1. The contribution of total Fe to $PM_{2.5}$ was the largest in dust days (5.2%), followed by rain (2.8%), clear (2.2%), fog (1.8%), and haze (0.8%) days. However, the contribution of dissolved Fe to $PM_{2.5}$ was the highest in fog days (0.11%), followed by dust (0.10%), haze (0.03%), clear (0.03%), and rain (0.02%) days.

Fe solubility in aerosols was calculated as dissolved /total Fe concentration $\times$ 100%. The average Fe solubility was the largest in fog days ($6.3 \pm 2.6$%), which was about 1.4, 3.0, 3.7, and 7.9 times higher than that in haze days ($4.6 \pm 1.9$%), dust days ($2.1 \pm 0.7$%), clear days ($1.7 \pm 0.6$%), and rain days ($0.8 \pm 0.3$%) (Fig. 2c). Although the concentration of total Fe in





dust days was the highest, Fe solubility was lower than that in fog and haze days. Fe solubility was extremely low in rain days, likely due to the removal of aged aerosols by wet deposition. Fe solubility differed significantly according to weather conditions ($p < 0.01$ or $0.05$). Compared with the data reported by Shi et al. (2020), Fe solubility in collected $PM_{2.5}$ samples during haze, fog, dust, and clear days reported in this study were all higher than those in total suspended particles (TSP) in Qingdao, a coastal city of China (haze: 1.75%; fog: 5.81%; dust: 0.27%; clear: 1.11%). This is not surprising considering the difference in the size cut of the aerosol collected. Other factors, such as different meteorological conditions and aerosol sources could also contribute to this difference.

### 3.3   Factors influencing Fe solubility

### 3.3.2 Sources of dissolved and total Fe

The primary sources of dissolved Fe are one of key factors that influencing Fe solubility. Pearson correlation analysis was employed to explore the primary sources of dissolved Fe under haze, fog, dust, and clear conditions (Table 2). Due to the influence of wet deposition, the data in rain days was not studied for Pearson correlations.

Previous studies reported that Al, Ca, and Ti are tracer elements for dust sources (Marsden et al., 2019; Buck et al., 2019). Se and As are tracers for the coal burning sources (Cui et al., 2019). Rai et al. (2020), Cai et al. (2017), and Chang et al. (2018) reported that Pb and Fe are mainly derived from steel industry or smelter. Liu et al. (2019) suggested that Co originate from the metal industry sources, and Cr is widely used in electroplating and leather industries. Zn is released from tires and motor oil and also from the use of motor vehicle brakes (Alias et al., 2020). Cu is mainly used in lubricants and in friction materials that constitute major contents of brake linings (Lin et al., 2015). K is often used as a tracer for biomass burning aerosol, but there are other sources such as dust (Bi et al., 2011).

In haze days, dissolved Fe had high correlations with Pb ($p < 0.01$), K ($p < 0.01$), and moderate correlations with Ca ($p < 0.05$), Ti ($p < 0.05$), Se ($p < 0.01$), Cr ($p < 0.01$), Zn ($p < 0.01$), Cu ($p < 0.01$). Pb, K, Se, Cr, Zn, and Cu had EF > 10, indicating the anthropogenic origin. The EF values of Ca and Ti were less than 10, suggesting the crustal sources. These results suggested a potential contribution of coal combustion, industrial emission, traffic emission, biomass burning, and dust to dissolved Fe under haze conditions.

In fog days, dissolved Fe showed moderate correlations with Al ($p < 0.01$), Ti ($p < 0.01$), Se ($p < 0.01$), As ($p < 0.01$), Cr ($p < 0.01$), Pb ($p < 0.05$), Zn ($p < 0.01$), and K ($p < 0.01$). Furthermore, the EF value of Ti was less than 10, while Se, As, Cr, Pb, Zn, and K had EF > 10. Similarly, these results indicated a potential contribution of coal combustion, industrial emission, traffic emission, biomass burning, and dust to dissolved Fe under fog conditions.



In dust days, dissolved Fe showed high correlations with Ca ($p < 0.01$), and moderate correlations with Al ($p < 0.05$), Ti ($p < 0.05$), Pb ($p < 0.05$), Zn ($p < 0.05$), and Cu ($p < 0.05$). The EF values of Ca and Ti were less than 10, while the EF values of Pb, Zn and Cu were larger than 10. Therefore, dust, industrial emission, and traffic emission may have contributed to dissolved Fe under dust conditions.

In clear days, dissolved Fe had moderate correlations with Al ($p < 0.01$), Ca ($p < 0.01$), Ti ($p < 0.01$), Se ($p < 0.05$), As ($p < 0.01$), Cr ($p < 0.05$), Zn ($p < 0.01$), and K ($p < 0.01$). The EF values of Ca and Ti were less than 10, while Se, As, Cr, Zn, and K had EF > 10. Therefore, coal combustion, industrial emission, traffic emission, biomass burning, and dust appear to be the dominant primary sources for dissolved Fe under clear conditions.

In order to further identify sources of dissolved Fe, a PMF model was used to apportion the sources of dissolved Fe. Figure S2, S3, and S4 show that 5, 6, and 7 factor profiles from the PMF model, respectively. Supplementary materials provided details on why 6 factors were selected as the final solution, and how the source of each factor was identified. PMF results showed that the main sources of dissolved Fe included dust, two types of industrial emissions, secondary sources, coal combustion, and traffic emission. The differences in the factor profiles and time series (Fig. S5) supported the split of the
two types of industrial emissions.

As shown in Figure 3, the contribution of dust, industrial emission 1, secondary sources, coal combustion, industrial emission 2, and traffic emission to dissolved Fe was 3.5%, 10.2%, 14.2%, 4.4%, 40.6%, and 27.1% in haze days, 2.5%, 8.1%, 16.3%, 3.4%, 42.6%, and 27.1% in fog days, 29.8%, 18.9%, 3.1%, 3.7%, 19.7%, and 24.8% in dust days, and 4.6%, 14.6%,
4.6%, 16.8%, 38.9%, and 20.5% in clear days. However, the contribution of dust, industrial emission 1, secondary sources, coal combustion, industrial emission 2, and traffic emission to total Fe was 25.6%, 12.4%, 14.6%, 1.5%, 12.3%, and 33.6% in haze days, 20.1%, 10.6%, 18.1%, 1.3%, 13.8%, and 36.1% in fog days, 77.3%, 8.1%, 1.1%, 0.5%, 2.1%, and 10.9% in dust days, 33.9%, 17.9%, 4.8%, 6.1%, 11.8%, and 25.5% in clear days. These results illustrated significant variations in the source apportionments during different weather conditions.


Although the source types were the same, the contributions of these sources to dissolved Fe were different from total Fe. For example, industrial emission (industrial emission 1 + industrial emission 2) was the largest contributor to dissolved Fe in haze (50.8%), fog (50.7%), dust (38.6%), and clear (53.5%) days, while it was the second or third largest contributor to total Fe in haze (24.7%), fog (24.4%), dust (10.2%), and clear (29.7%) days. Traffic emission was the second largest contributor
to dissolved Fe in haze (27.1%), fog (27.1%), dust (24.8%), and clear (20.5%) days, while it was the first or second largest contributor to total Fe in haze (33.6%), fog (36.1%), dust (10.9%), and clear (25.5%) days. In addition, the PMF results also revealed that secondary sources was a significant contributor to dissolved Fe in haze (14.2%) and fog (16.3%) days, although it was less important in dust (3.1%) and clear (4.6%) days. Note that due to the limitation of PMF, it is difficult to fully



separate secondary sources of dissolved Fe (i.e., solubilised from insoluble Fe due to atmospheric processing) from primary
sources. This means that some of dissolved Fe due to atmospheric processing may still be assigned to its primary factors if
there is a strong co-variation between dissolved Fe and primary tracers. Therefore, the contribution of secondary sources to
dissolved Fe is likely higher than that indicated by the PMF. In the following, we further investigated the mixing of acidic
species and Fe aerosols in haze and fog days to provide further evidence of the Fe solubilisation from primary insoluble Fe
aerosols.

**3.3.2  Atmospheric acidification processing**

A number of studies have considered atmospheric acidification processing as a key factor influencing Fe solubility, in
addition to direct emission of dissolved Fe from primary sources (Ito and Shi, 2016; Li et al., 2017; Zhang et al., 2019a; Shi
et al., 2020; Liu et al., 2021). As mentioned above, a significant proportion of dissolved Fe was associated with a PMF factor
identified as secondary source during haze and fog days, suggesting a significant contribution from atmospheric processing.
To further support this result, a total of 688 and 404 individual particles were analyzed by TEM/EDS in haze and fog days,
respectively. TEM analysis showed two types of Fe-containing particles: Fe-rich and S-Fe particles. Figure 4 shows that Fe-
rich particles usually contain aggregates of multiple spherical Fe particles. TEM/EDS also detected minor Fe besides major
elements (S, C, and O) in acidic secondary aerosols, and named as S-Fe particles (Fig. 4). This is similar to that reported by
Li et al. (2017) who then confirmed that such Fe was presented as Fe sulfate from nano-scale secondary ion mass
spectrometry (NanoSIMS) observations, indicative of acid dissolution.

We further calculated the number contribution of Fe-containing particles to all analyzed particles: 7.4% in haze days and 6.7%
in fog days. The number contribution of S-Fe particles to Fe-containing particles was 76.3% and 87.1% in haze and fog days,
respectively, suggesting that Fe particles were mostly internally mixed with acidic aerosol species under haze and fog days.
In addition, Figure 5 shows that acidic secondary aerosol species (e.g., sulfate and nitrate) increase the size of Fe particles by
about 3.6 and 2.4 times under haze and fog days, respectively.

As mentioned above, total Fe concentration in fog days was higher than that in haze days (Fig. 2a), whereas sulfate and
nitrate did not display high concentrations in fog days compared with haze days (Fig. S6). Primary sources of dissolved Fe in
haze and fog days were similar. However, Fe solubility in fog days was much higher than that in haze days.

Under fog condition, RH ranged from 71% to 99%. This is much higher than the threshold (50%) of the particle surface
changed to wet or liquid state (Sun et al., 2018). The aqueous phase makes it easier to take up acidic gases (such as $SO_2$, $NO_2$)
to produce acidic salts (such as sulfate, nitrate), which in turn enable the particles to absorb more water vapor (Shi et al.,
2020). The acids can then promote Fe dissolution (Li et al., 2017). Furthermore, dissolved Fe may catalyze the oxidation of
acidic gases to acidic salts (Wang et al., 2019), which further enhances Fe dissolution.





To further support this argument, we used the molar ratio of acidic ions to total Fe to investigate the impact of aerosol acidification on Fe solubility. The method was proposed by Hsu et al. (2014) and had been used by Shi et al. (2020) and Zhu et al. (2020). In this study, $SO_4^{2-}$ and $NO_3^-$ were used to indicate aerosol acidification, because these two ions were the predominant acidic components in $PM_{2.5}$ (Fig. S6). Here, we calculated aerosol acidification as $(2SO_4^{2-} + NO_3^-)$/total Fe molar ratio (Figure 6). The correlation between $(2SO_4^{2-} + NO_3^-)$/total Fe molar ratio and Fe solubility in fog days (r = 0.78, p < 0.01) was higher than that in haze days (r = 0.61, p < 0.01). Moreover, the slope of the regression line for fog condition was 0.12, while it was 0.10 for haze condition (Fig. 6). This further supported the above argument that the solubilisation of Fe aerosols by acids was more effective under fog conditions.

In haze days, RH ranged from 41%-79%. When RH > 50%, average $(2SO_4^{2-} + NO_3^-)$/total Fe molar ratio was 35.7 μmol μmol⁻¹ in haze days, which was higher than that in fog days (31.4 μmol μmol⁻¹). However, Fe solubility in haze days was somewhat lower than that in fog days (5.6% vs. 6.3%). This could be due to the low RH in haze days, which led to lower water content on the particles relative to fog days. The low water content in the aerosol particles may have limited the uptake and oxidation of acidic gases. When RH < 50%, Fe solubility in haze days was lower than 3.6%, even if $(2SO_4^{2-} + NO_3^-)$/total Fe molar ratio was high.

## 4  Summary and atmospheric implications

The average Fe solubility was the largest in fog days (6.3 ± 2.6%), which was about 1.4 times higher than that in haze days (4.6 ± 1.9%), 3.0 times higher than that in dust days (2.1 ± 0.7%), 3.7 times higher than that in clear days (1.7 ± 0.6%), and 7.9 times higher than that in rain days (0.8 ± 0.3%). Although small in dust (3.1%) and clear (4.6%) days, secondary source significantly contributed to dissolved Fe in haze (14.2%) and fog (16.3%) days. Individual particle analysis further showed that about 76% and 87% of Fe-containing particles were internally mixed with sulfates and nitrates under haze and fog conditions, respectively. Our study indicated that aqueous surface of aerosol particles (when RH > 50%) may facilitate the update of acidic species and thereby promote Fe dissolution and increase Fe solubility. Higher RH in fog days (71%~99%) compared with haze days (41%~79%) resulted in more effective aerosol acidification and higher Fe solubility.

Maher et al. (2016) and Lu et al. (2020) reported that when atmospheric $Fe_3O_4$ particle with size < 200 nm can access the brain directly via transport through the neuronal axons of the olfactory or trigeminal nerves. In this study, the peak sizes of Fe-rich particles were 175 nm, 200 nm, 225 nm, and 175 nm in haze, fog, dust, and clear days, respectively. Dissolved Fe may also induce oxidative stress (Abbaspour et al., 2014). Therefore, Fe aerosols, regardless of the weather conditions, are a potential hazard to human health in densely populated megacities.

Fe-containing particles from the continent can be transported and further deposited to the ocean (Winton et al., 2015;
Yoshida et al., 2018; Conway et al., 2019). Li et al. (2017) found large amounts of anthropogenic fine Fe-containing
particles in the East China Sea. In this study, the prevailing winds during the sampling period dominated by the west or
northwest winds under haze, fog, and dust conditions, suggesting that Fe-containing particles were likely transported into the
ocean. We also found that the significant contribution of dissolved Fe from atmospheric heterogeneous reactions (i.e.,
aerosol acidification) should be paid attention in haze and fog days.

**Data availability**

The data used in this study are available from the corresponding author upon request (email: liweijun@zju.edu.cn).

**Author contributions**

YHZ, WJL, and ZBS designed the study. JZ, YHZ, LL, and LX collected aerosol and individual particle samples. YHZ and
YW contributed laboratory experiments. YHZ, WJL, ZBS, and JSX performed data analysis. YHZ and WJL wrote the paper
and prepared the manuscript material with contributions from all the co-authors. JHS, LYS, PQF, DZZ and ZBS commented
the paper.

**Competing interests**

The authors declare that they have no conflict of interest.

**Acknowledgements**

We thank Atmospheric Science Practice Center of School of Earth Sciences, Zhejiang University for sharing meteorological
data during the sampling period. This work was funded by the National Natural Science Foundation of China (41907186,
42075096), China Postdoctoral Science Foundation (2019M652059), Zhejiang Provincial Natural Science Foundation of
China (LZ19D050001). Zongbo Shi acknowledges funding from the UK Natural Environment Research Council
(NE/N007190/1 and NE/R005281/1).

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



**Tables**

Table 1. Percentage contributions of total and dissolved Fe concentrations to PM$_{2.5}$ concentration under haze, fog, dust, clear, and rain conditions. The maximum and minimum values are in brackets.

|  | **Haze** | **Fog** | **Dust** | **Clear** | **Rain** |
|---|---|---|---|---|---|
| Total Fe/PM$_{2.5}$ | 0.8 ±0.4 | 1.8 ±1.2 | 5.2 ±1.9 | 2.2 ±0.9 | 2.8 ±1.6 |
|  | (0.4–2.2) | (0.7–5.9) | (3.3–10.7) | (0.8–4.4) | (1.1–6.3) |
| Dissolved Fe/PM$_{2.5}$ | 0.03 ±0.01 | 0.11 ±0.09 | 0.10 ±0.02 | 0.03 ±0.02 | 0.02 ±0.01 |
|  | (0.00–0.07) | (0.04–0.38) | (0.07–0.13) | (0.01–0.13) | (0.00–0.05) |

Table 2. Pearson correlation coefficients (r) between dissolved Fe and trace elements. The Pearson correlation coefficient |r| < 0.40, 0.40 < |r| < 0.70, and |r| > 0.70 reflects a low, moderate, and high correlation, respectively.

|  | **Haze (n = 30)** | **Fog (n = 28)** | **Dust (n = 12)** | **Clear (n = 30)** |
|---|---|---|---|---|
| Al | -0.27 | 0.58$^{**}$ | 0.61$^{*}$ | 0.48$^{**}$ |
| Ca | 0.41$^{*}$ | 0.36 | 0.76$^{**}$ | 0.62$^{**}$ |
| Ti | 0.44$^{*}$ | 0.68$^{**}$ | 0.66$^{*}$ | 0.64$^{**}$ |
| Se | 0.68$^{**}$ | 0.69$^{**}$ | 0.22 | 0.41$^{*}$ |
| As | -0.23 | 0.56$^{**}$ | 0.35 | 0.47$^{**}$ |
| Cr | 0.58$^{**}$ | 0.59$^{**}$ | -0.09 | 0.46$^{*}$ |
| Pb | 0.73$^{**}$ | 0.47$^{*}$ | 0.66$^{*}$ | 0.25 |
| Co | -0.17 | 0.34 | 0.43 | -0.31 |
| Zn | 0.47$^{**}$ | 0.51$^{**}$ | 0.46$^{*}$ | 0.46$^{**}$ |
| Cu | 0.51$^{**}$ | 0.34 | 0.62$^{*}$ | 0.32 |
| K | 0.76$^{**}$ | 0.49$^{**}$ | -0.25 | 0.52$^{**}$ |

$^{*}$. Correlation is significant at the 0.05 level (2-tailed).

$^{**}$. Correlation is significant at the 0.01 level (2-tailed).




**Figures**

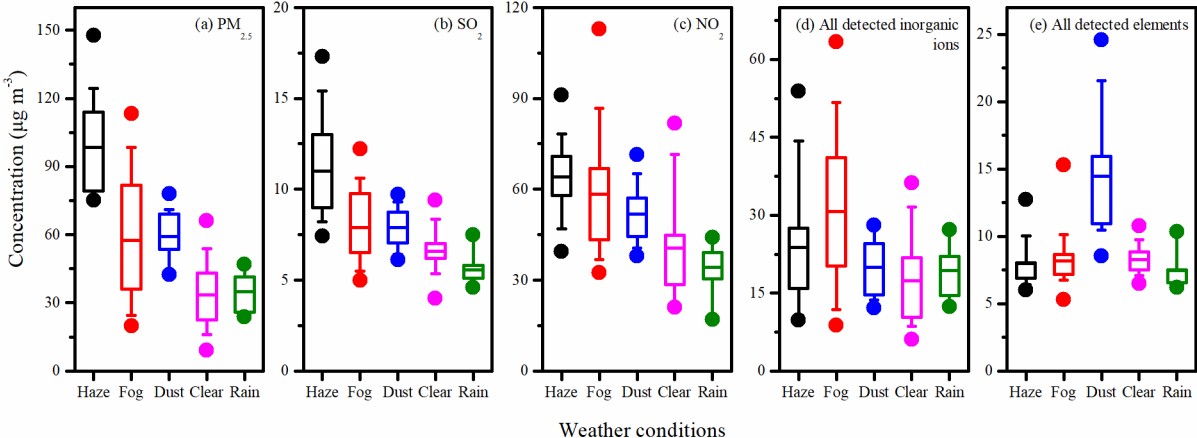

**Figure 1.** PM$_{2.5}$ (a), SO$_2$ (b), NO$_2$ (c), all detected inorganic ions (d), and all detected elements (e) concentrations under haze, fog, dust, clear, and rain conditions. The solid circles above and below the box show the maximum and minimum values, respectively. The whiskers above and below the box indicate the 90th and 10th percentiles, respectively. The boundaries of the box represent the 25th and 75th percentiles. The solid line inside the box indicates the mean value.

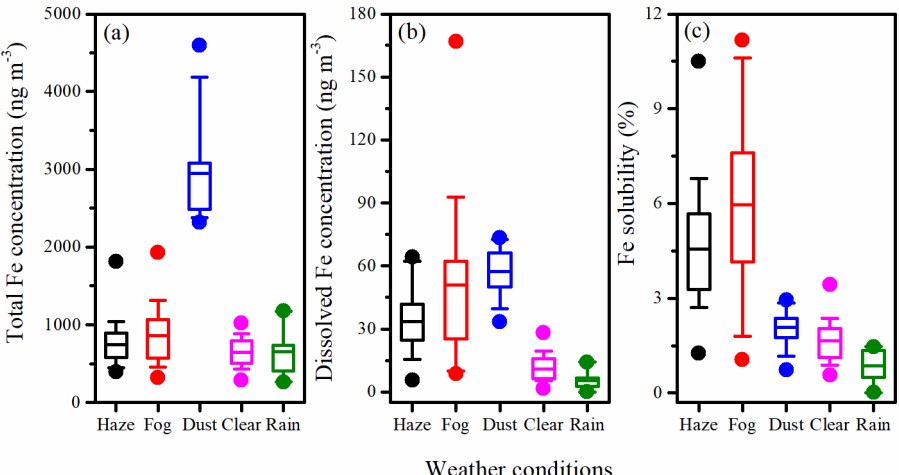

**Figure 2.** The box and whisker plot of the concentrations of total Fe (a) and dissolved Fe (b) as well as Fe solubility (c) under haze, fog, dust, clear, and rain conditions. The solid circles above and below the box show the maximum and minimum values, respectively.






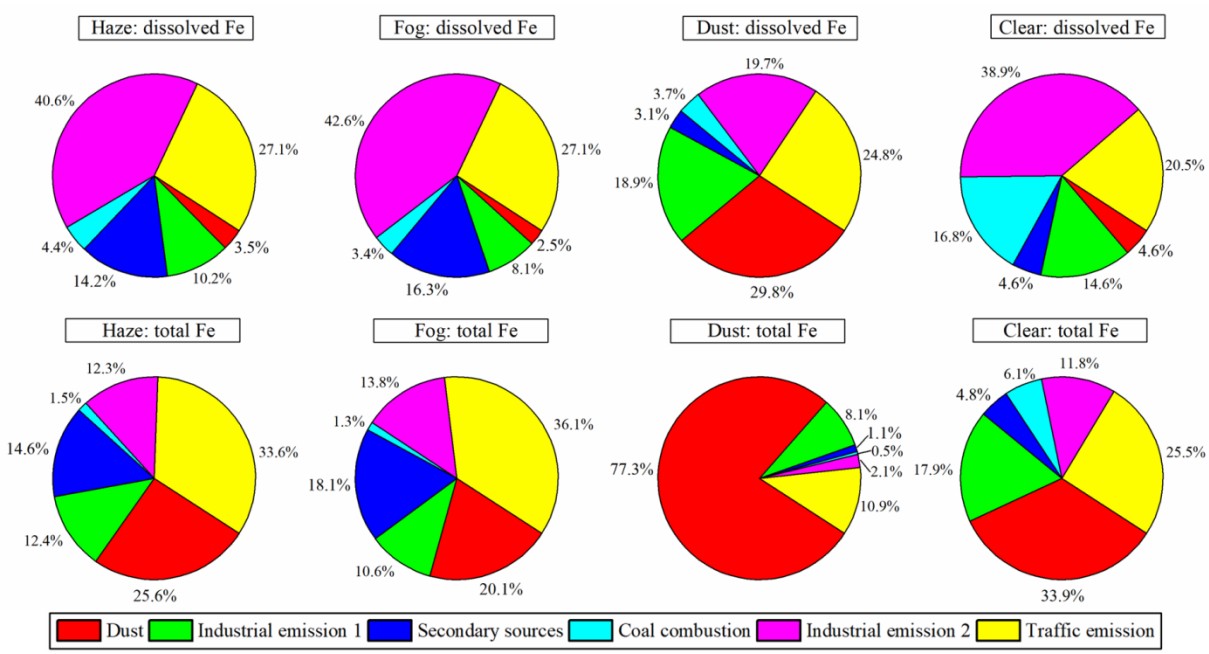

**Figure 3. Contributions of identified sources for dissolved and total Fe in haze, fog, dust, and clear days by the PMF model.**


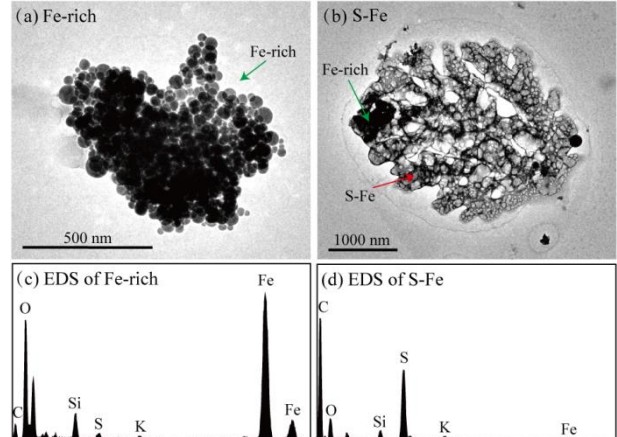

**Figure 4. Typical TEM images and corresponding EDS spectra of Fe-rich and S-Fe particles: (a) TEM image of Fe-rich particle, (b) TEM image of S-Fe particle, (c) EDS of Fe-rich particle, (d) EDS of S-Fe particle.**





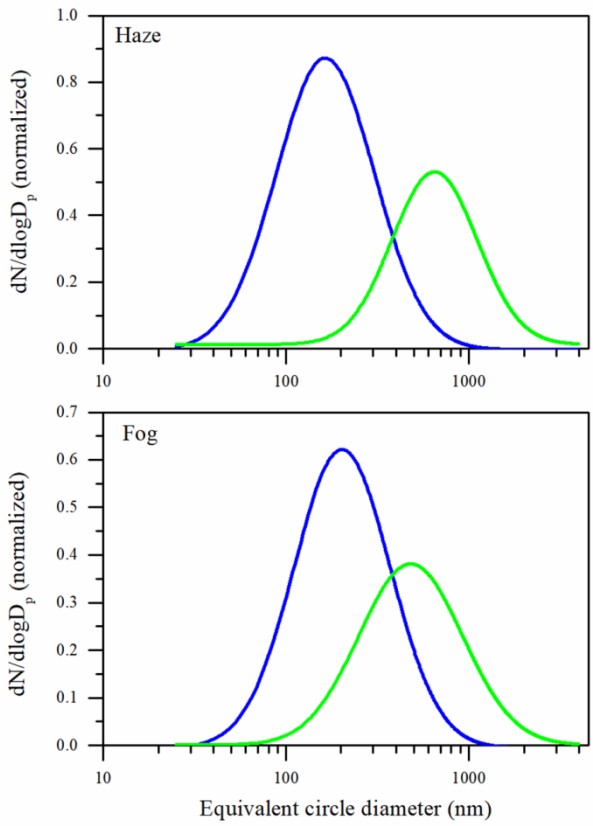

**Figure 5. Size distributions of Fe-rich (blue line) and S-Fe (green line) particles under haze and fog days. The distribution pattern is normalized.**

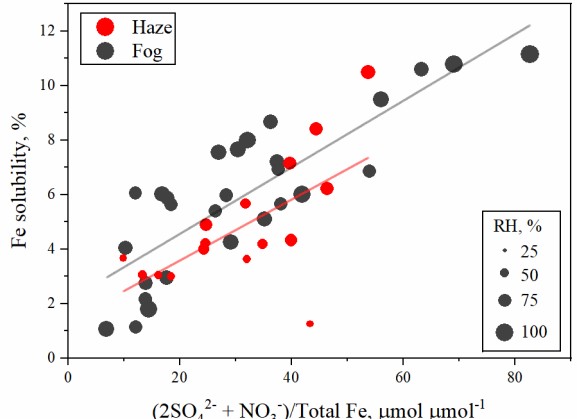

**Figure 6. Correlations between Fe solubility and $(2SO_4^{2-} + NO_3^-)$/total Fe molar ratio under different RH. The linear regression: y = 0.12x + 2.12, r = 0.78, p < 0.01 (fog); y = 0.10x + 1.57, r = 0.61, p < 0.01 (haze).**