# Peer review of "Sources and processes of iron aerosols in a megacity of Eastern China"

_Atmospheric Chemistry and Physics, 2021_

## Author Comment (AC1)

**Responses to the reviewer comments on**

**"Sources and processes of iron aerosols in a megacity of Eastern China" by Zhu et al.**

The authors would like to thank both reviewers for their constructive and good suggestions to improve our manuscript. We have carefully considered all the review comments and revised the manuscript. Below, we provide responses to the comments in blue, with changes made in the manuscript highlighted in red.

**Response to Reviewer 2:**

*This paper deals with the Fe aerosol particles reactivity in an urban environment, according to variable weather conditions. I would like to commend the authors for their work: it is a very relevant study, dealing with an essential issue in Fe atmospheric chemistry for four main reasons: Soluble Fe plays an important role in many environmental processes, including in ocean biogeochemistry and thereby the global carbon cycle; Fe-bearing particles may have adverse health effects; Anthropogenic Fe particles have been the subject of increased interest in recent years due to their significant solubility; Very few atmospheric Fe solubilities inferred from urban field studies have been reported in the literature. I cannot however recommend the publication of this study in a high ranked journal as Atmospheric Chemistry and Physics in its present form. Please find below some suggestions, as an attempt to improve the manuscript before resubmission.*

Response: We appreciate the reviewer for the helpful comments and suggestions. Below we address the comments and have revised the manuscript accordingly. For clarity, the reviewer's comments are listed below in *black italics*, while our responses and changes in manuscript are shown in blue and red, respectively.

*1. Section 3.1 (Pollution Levels): This section only gives an overview of the air pollution in the study area, without the results being directly related to the rest of the study, i.e. the evolution of the solubility of particulate iron as a function of the ambient meteorological conditions. I therefore suggest that the authors place the*

*detailed discussion of the results of this section in the "Supplementary Information" section and keep only a summary in the main body of the text.*

Response: We have moved descriptions about $SO_2$, $NO_2$, inorganic ions and elements to the Supplemental Information, and the changed Section 3.1 are as follows:

The average $PM_{2.5}$ concentration was the highest at $98.3 \pm 20.6$ μg m$^{-3}$ in haze days, followed by $59.3 \pm 11.1$ μg m$^{-3}$ in dust days, $57.5 \pm 26.9$ μg m$^{-3}$ in fog days, $33.6 \pm 14.5$ μg m$^{-3}$ in clear days, and $31.4 \pm 8.1$ μg m$^{-3}$ in rain days (Fig. S2). About 100%, 29%, and 8% of $PM_{2.5}$ concentrations in haze, fog, and dust days were higher than the Grade Ⅱ national $PM_{2.5}$ standard of 75 μg m$^{-3}$ (24 h average standard, GB 3095-2012, China), respectively. However, all of $PM_{2.5}$ concentrations in clear and rain days were lower than the $PM_{2.5}$ Grade Ⅱ standard. $PM_{2.5}$ concentrations differed significantly according to weather conditions ($p < 0.01$, independent sample T test, Table S4).

The concentrations of $SO_2$, $NO_2$, all detected inorganic ions and elements also differed significantly according to weather conditions (Table S4). The concentration order of $SO_2$ or $NO_2$ in different weather conditions was haze > fog > dust > clear > rain days (Fig. S2). However, the concentration orders of all detected inorganic ions and elements were fog > haze > dust > rain > clear days and dust > clear > fog > haze > rain days, respectively. Detailed descriptions of $SO_2$, $NO_2$, all detected inorganic ions and elements were given in Supplemental Information.

(Page 7, Line 184-195)

*2. Section 3.2 (Fe content and solubility): At the end of this section the authors compare their results (PM2.5) with those of Shi et al. (2020) for TSP and state that it is not surprising that the solubilities reported in the present study are consistently higher than those reported by Shi et al. This statement seems premature to me because, to my knowledge, Shi et al. do not provide any indication of what the coarse fraction of the aerosol (> 2.5 microns) in their samples represents. In my opinion, the end of section 3.2 should be deleted as it does not add anything to the authors' statements.*

Response: We agree. We deleted the comparison with Shi et al. (2020).

*3. Section 4. Summary and atmospheric implications: It is surprising that the authors*

*can state that the majority of the iron particles observed in TEM-EDS contain, in addition to sulfates, nitrate ions, because nitrogen is extremely difficult to detect in individual analysis by this technique, unless a cryogenic system is available. Moreover, no nitrogen signal is visible on the spectra of Figure 4. What is the basis for the authors' assertion that iron is associated with nitrates in the collected aerosol?*

Response: It is true that N peak is very low or hard to be seen in Figure 4. Our previous individual particle analyses, including cryogenic TEM, have clearly shown that individual particles in urban air more or less contain sulfate, nitrate and secondary organic matter (Li et al., 2016). This has been confirmed in single particle mass spectrometry studies (Whiteaker et al., 2002). We have added corresponding sentences in the manuscript as follows:

It should be noted that individual secondary sulfate particle in urban air normally contain nitrate, which has been confirmed in single particle mass spectrometry studies (Whiteaker et al., 2002; Li et al., 2016).

(Page 12, Line 361-363)

References:

Li, W., Sun, J., Xu, L., Shi, Z., Riemer, N., Sun, Y., Fu, P., Zhang, J., Lin, Y., and Wang, X.: A conceptual framework for mixing structures in individual aerosol particles, J. Geophys. Res. Atmos., 121, 13784-13798, https://doi.org/10.1002/2016JD025252, 2016.

Whiteaker, J. R., Suess, D. T., and Prather, K. A.: Effects of Meteorological Conditions on Aerosol Composition and Mixing State in Bakersfield, CA, Environ. Sci. Technol., 36, 2345-2353, https://doi.org/10.1021/es011381z, 2002.

*4. Section 2.2: Sample collection: When collecting aerosol samples during rain or fog days, there is a risk that the surface of the filter will be washed away and that leaching of the particles will occur. Thus the soluble fraction of the aerosol will be carried into the air pumping system. What precautions do the authors take to avoid this leaching?*

Response: Particulate matter (PM) samplers are designed to be water proof, so no water will get into the samplers to wet the filters even under heavy rain and storm.

Moreover, we collected PM samples with a PM$_{2.5}$ sampling head, rather than a total suspended particulate (TSP) inlet. A majority of cloud and fog droplets are larger than 2.5 μm, so they are not collected into our samplers. It is possible that some tiny fog / cloud droplets have been collected, but the large surface area and small mass of particles mean that such fog or cloud droplets will not cause the "leaching" as mentioned by the reviewer. This is further confirmed in the visual inspection of the filters after sampling. Therefore, we are highly confident that the "leaching" effect does not exist in our samples.

*5. Section 3.3.2: Atmospheric acidification processing, lines 293-294: The authors state that the fact that a significant proportion of dissolved iron is associated with secondary sources is evidence of the important contribution of atmospheric processing to soluble iron production. I am absolutely convinced of the importance of atmospheric processes in the production of soluble iron. However, examination of Figure 3 indicates that industrial type 2 sources contribute equally to soluble iron production regardless of weather conditions (38.9 to 42.6%, except for dusty days). This demonstrates to me that the chemical composition of particulate matter emitted by industry is as important as atmospheric processes in the production of soluble iron. I would therefore suggest that the authors be careful when they insist on the influence of atmospheric processes in the production of soluble iron.*

Response: We appreciate your comments. We checked and re-calculated PMF results, and found that industrial emissions were still the largest contributor to dissolved Fe. The revised Figure 3 is as follows:

[Figure]

Figure 3. Contributions of identified sources to dissolved Fe, total Fe, and PM$_{2.5}$ in haze, fog, dust, and clear days by PMF model.

We added discussions about the large contributions of industrial emissions 1 & 2 and the relatively low contributions of secondary sources to dissolved Fe in the manuscript as follows:

Figure 3 also shows that although industrial emissions (factor 5&6 or industrial emissions 1 + industrial emissions 2) contributed less than 20% to PM$_{2.5}$ in haze, fog, dust, and clear days, they were the largest contributor to dissolved Fe in haze (65.4%), fog (72.4%), dust (44.5%), and clear (62.5%) days, and they also were the largest contributor to total Fe in haze (44.2%), fog (55.0%), and clear (39.1%) days except dust days. Industrial emissions 1 (factor 5) contributed similarly to dissolved Fe regardless of weather conditions (38.9% to 43.6%, except for dusty days), while it only contributed 11.6% to 13.9% to total Fe (except dusty days). Heavy oil

combustion related aerosols have the highest Fe solubility (up to 78%) from all major Fe aerosol sources (Schroth et al., 2009; Ito et al., 2021). This may explain the much larger contribution of industrial emissions 1 to dissolved Fe than total Fe. As far as we know, there is no published data on Fe solubility in particulate matter from metal industrial emissions. Considering the dominance of iron and steel plants in total Fe emissions (Rathod et al., 2020) and the low Fe solubility in smelter ash from a steel plant (Li et al., 2017), it is difficult to understand why industrial emissions 2 (factor 6) contributes so much to dissolved Fe. Furthermore, PMF results indicated that secondary sources were the largest contributor to $PM_{2.5}$ in haze (66.2%), fog (72.3%), and clear (31.2%) days except dust days. However, the contribution of secondary sources to dissolved Fe was relatively low: 16.1% in haze days, 16.5% in fog days, 3.1% in dust days, and 5.4% in clear days.

The likely reason for the high contribution of industrial emissions 2 and the relatively low contribution of secondary sources to dissolved Fe is that PMF is unable to completely separate secondary sources of dissolved Fe (i.e., dissolved from insoluble Fe due to atmospheric processing) from primary sources. This means that some of dissolved Fe due to atmospheric processing may still be assigned to its primary factors if there is a strong co-variation between dissolved Fe and primary tracers. This suggests that the contribution of secondary sources to dissolved Fe is likely higher than that indicated by the PMF. It should also be noted that industrial emissions are outside the city and thus particles from these sources undergo long-range transport before reaching the sampling site. This provides more time for chemical processing in the atmosphere, leading to Fe solubilisation.

(Page 11, Line 325-347)

---

## Author Comment (AC2)

**Responses to the reviewer comments on**

**"Sources and processes of iron aerosols in a megacity of Eastern China" by Zhu et al.**

The authors would like to thank both reviewers for their constructive and good suggestions to improve our manuscript. We have carefully considered all the review comments and revised the manuscript. Below, we provide responses to the comments in blue, with changes made in the manuscript highlighted in red.

**Response to Reviewer 1:**

*This paper appears to be a data paper with litter data analysis of the beyond reporting the results. Although the authors try to explain the sources or atmospheric phenomena that lead to the changes of iron aerosols under different weather conditions, the discussion are too general. Additionally, the English language requires substantial improvement (both style and grammar) throughout the manuscript. Many sentences are not clearly written. The topic is certainly appropriate for Atmospheric Chemistry and Physics. However, there are quite a few major issues with the study that prevent me from recommending it for publication in the present format. It is possible that these issues could be addressed with a major revision. My specific concerns are addressed below.*

Response: We thank reviewer#1 for the helpful comments. Below, we address the comments and have revised the manuscript accordingly. We have significantly enhanced the discussions, including a detailed discussion on the assignment of factors. We revised the English language accordingly. For clarity, the reviewer's comments are listed below in *black italics,* whereas our responses and changes in manuscript are shown in blue and red, respectively.

*1. Line 43: Change "having" to "have".*

Response: We have changed "having" to "have" as follows:

Although natural emissions have a high emission flux, their contribution to Fe solubility is less than 1% (Schroth et al., 2009).

*2. Line 49: Fe solubility in some anthropogenic sources such as coal fly is very low, so here the statement "an increase … from anthropogenic source cloud lead to the increase in Fe solubility" is not accurate.*

Response: We have changed the corresponding sentence as follows:

These results imply that an increase in relative amounts of aerosols from these mixed anthropogenic sources may be responsible for the increase in Fe solubility.

*3. Line 55: When the relative humidity is higher than 50%, some soluble inorganic components may begin to be hygroscopic, resulting in phase changes of particles, but some secondary organic components will not. For some aged aerosol particles after liquid-liquid phase separation, the organic coating also prevents inorganic components from contacting the atmosphere directly, which would affect the hygroscopic property of particles. Moreover, in the reference cited here, hygroscopic growth begins at 60% and 55% for haze particles. Therefore, the "50%" or "secondary aerosol particles" here are not appropriate, please consider rewording them. Additionally, what do the surfaces of secondary aerosol particles mean? You mean the secondary materials coat the primary particles, or the particles are secondarily formed? Please clarify.*

Response: We fully agree with the reviewer's comments. When RH is higher than 60%, the surface of aerosol particles will change to wet or liquid state. Recently, one of our studies shows that more than half of secondary inorganic particles are not coated by organic coating (Li et al., 2021). The organic coating in liquid-liquid phase separated particles can prevent the water uptake. However, in this paper, we do not plan to discuss this in great detail. But to make it clear, we revised the corresponding sentences as follows:

When ambient RH is above 60%, aerosol particles can take up water and change the surface to wet or liquid state (with liquid-liquid separation or homogenous, depending on the composition and RH) (Sun et al., 2018; Liu et al., 2017).

Reference:

Li, W., Teng, X., Chen, X., Liu, L., Xu, L., Zhang, J., Wang, Y., Zhang, Y., and Shi, Z., Organic Coating Reduces Hygroscopic Growth of Phase-Separated Aerosol Particles, Environ. Sci. Technol., https://pubs.acs.org/doi/10.1021/acs.est.1c05901, 2021.

We also checked the threshold of particle surface changed to wet or liquid state throughout the manuscript as follows:

Under fog condition, RH was higher than 90%, which was much higher than the threshold (60%) of the particle surface changed to wet or liquid state (Sun et al., 2018; Liu et al., 2017).

(Page 13, Line 385-386)

When RH > 60%, average aerosol acidity/total Fe was 2.3 μmol μmol$^{-1}$ and 2.1 μmol μmol$^{-1}$ in haze and clear days, respectively, which were similar with that in fog days (2.4 μmol μmol$^{-1}$).

(Page 13, Line 387-389)

When RH < 60%, Fe solubility in haze and clear days was lower than 3.9% and 2.3%, respectively, even when aerosol acidity/total Fe was high.

(Page 13, Line 392-393)

Our study indicated that wet surface of aerosol particles (when RH > 60%) may facilitate the update of acidic species and thereby promote Fe dissolution and increase Fe solubility.

(Page 13, Line 403-405)

[Figure]

Figure 5. Correlations between Fe solubility and aerosol acidity/total Fe under different RH.

(Page 26)

*4. Lines 62-64: It is not a complete sentence.*

Response: We have changed the sentence as follows:

The two major contributors mentioned above (aerosol primary sources and atmospheric acidification processes) to Fe solubility are associated with weather conditions, which can change dispersion efficiency (such as boundary layer height, wind, and convection), dry/wet deposition, and chemical conversion loss rate (Leibensperger et al., 2008; Zhang et al., 2018), temperature, relative humidity, and solar radiation (Camalier et al., 2007).

(Page 3, Line 64-68)

*5. Line 84: The identification of fog samples: as described in Table S2, the threshold value of 10 km is too high for fog visibility. If the relative humidity during fog is not considered in the definition of weather conditions, in my opinion, it will lead to the misclassification of the fog samples. The authors do not mention the RH in the definition of fog. Please clarify.*

Response: This was a typo. The visibility in fog day is less than 1 km instead of 10 km. The relative humidity is higher than 90% in fog day. We should emphasize that our classification is the same as the Chinese Meteorological Administration (CMA) reports. We changed the threshold of fog visibility, and added the threshold of relative humidity in Table S2 in Supplemental Information as follows:

**Table S2. Definitions of haze, fog, dust, clear, and rain weather conditions.**

| | Definition |
|---|---|
| Haze | The meteorological definition of haze is a kind of weather phenomenon in which a large number of tiny dust particles, smoke particles or salt particles suspended in the atmosphere, the relative humidity is less than 80%, and the horizontal visibility drops below 10 km. |
| Fog | The meteorological definition of fog is tiny water droplets suspended in the air, and horizontal visibility is less than 1 km, the relative humidity is higher than 90%. |
| Dust | Dust is a kind of natural meteorological phenomenon associated with strong cold front from Northwest China. The FLEXible PARTicle (FLEXPART) Lagrangian particle dispersion model shows that air mass backward trajectories of typical dust events crossed East Asia (Fig. S1). |
| Clear | Clear weather samples were collected when $PM_{2.5}$ concentration was less than 75 μg m$^{-3}$, and visibility was greater than 10 km. |
| Rain | Rain refers to the liquid droplets falling to the ground from the above cloud. We collected $PM_{2.5}$ samples as rain samples when precipitation intensity $< 10$ mm d$^{-1}$. |

*6. Line 89: The daytime and nighttime samples are collected respectively. It is not quite clear how the sampling strategy was selected, and why the authors do not discuss the differences between daytime and nighttime samples.*

Response: The number of samples in fog and dust days is only 8 and 6 in the day, and 9 and 6 in the night, respectively. Such a small number of daytime and nighttime samples in fog and dust days is not enough to obtain accurate results of source identification and correlation analysis between Fe solubility and aerosol acidity, and between Fe solubility and liquid water content. In order to maintain consistency throughout the manuscript, we did not discuss the differences between daytime and nighttime samples.

*7. Lines 88-103: The PM2.5 samples are used for the mass concentration analysis of total and dissolved Fe, but the particle size of the samples used for single particle analysis is not clearly indicated in this paper. Is there any difference in the cut point inlet between single-stage cascade impactor and TH-16A Intelligent sampler? The authors state that the collection efficiency is 50% for particles with an aerodynamic diameter of 0.1 μm and a density of 2 g cm-3, so the aerodynamic particle size of particles collected from single-stage cascade impactor is less than 100 nm? If so, I suspect that the collected particles have not yet grown.*

Response: The TH-16A Intelligent sampler can collect aerosol particles $< 2.5$ μm. The

collected PM$_{2.5}$ samples are used to analyze the mass concentrations of total Fe by Energy Dispersive X-Ray Fluorescence spectrometer and dissolved Fe by Ultraviolet-Visible spectrophotometer.

The single particle sampler can collect >100 nm aerosol particles with the collection efficiency at 50%. The statement *"the collection efficiency is 50% for particles with an aerodynamic diameter of 0.1 μm"* does not mean that the sampler could not collect ultrafine particles (< 100 nm) on the substrate. However, the collection efficiency of < 100 nm is much smaller than 50%. The collected single particle samples were used to analyze chemical composition, morphology characteristics, size distribution and mixing state of single particles by transmission electron microscope-energy-dispersive X-ray spectrometer (TEM-EDS).

Because TEM-EDS can measure the individual particle size, it is easy to know the size of the single particle being analyzed. Figure 6 shows that the size range of Fe-containing particles is 25-5000 nm. The different types of samplers are used to answer different questions – TH-16A for bulk composition, and single particle sampler for understanding the mixing state of individual particles.

*8. Line 155: A reference would be helpful.*

Response: As shown in comment 13, to avoid similar statements with positive matrix factorisation (PMF), the source results obtained by enrichment factor (EF) analysis were deleted. Therefore, references were not added.

*9. Line 226: Change "3.3.2" to "3.3.1".*

Response: We have changed "3.3.2" to "3.3.1".

(Page 8, Line 242)

*10. Lines 227-258: The authors applied Pearson correlation analysis between dissolved Fe and other elements to explore the primary of dissolved Fe. The elements do not consider the impact of atmospheric process, but the dissolved Fe is affected by the atmospheric process, so I am wondering whether the correlation analysis between the dissolved Fe and other elements can be used for source identification.*

Response: As mentioned in response to comment 13, the source results obtained by Pearson correlation analysis were deleted. Because PMF results can not only get the

source type, but also the contribution of each source, so the PMF result is retained.

*11. Lines 241-243, 246-248, 252-253 and 256-258: The authors state that EF values of Ca and Ti are less than 10, suggesting a potential contribution of coal combustion,...; Pb, Zn and K had EF > 10, indicating a potential contribution of coal combustion, ... These statements make no sense.*

Response: We agree with the reviewer's comments. We have deleted the source results based on EF values.

*12. Lines 260-265: I believe that the figures represented in supplement appear to be more important. While reading that authors have referred to supplementary figures too many times, I would suggest merging or re-plotting some of the figures to bring supplementary figures in main text.*

Response: We have migrated Figure S3 (source profiles deduced from PMF analysis (6 factors)) from supplement to manuscript. The descriptions about the sources represented by each factor and the explanations for why 6 factors were selected as final solution were also putted in the manuscript. Figure S2 (source profiles deduced from PMF analysis (5 factors)) and Figure S4 (source profiles deduced from PMF analysis (7 factors)) were still in the supplementary file.

*13. Lines 267-273: The Figure 3 has provided the contribution of each source in detail, so the authors do not need to describe them again. The similar situations appear many times in the text.*

Response: We have deleted the descriptions about Figure 3 in line 274-281. To avoid similar statements, the source results obtained by enrichment factor (EF) and Pearson correlation analysis were deleted. Now 
[revised manuscript text omitted]

(Page 9, Line 276-Page 12, Line 349)

*14. Lines 2677-283: Oakes et al. (2012) found that Fe solubility was 0.06% in coal fly, 46% in biomass burning, 51% in diesel exhaust and 75% in gasoline exhaust as the authors state in Lines 47-48. Here the industrial emission is the largest contributor to dissolved Fe in haze, fog, dust and clear days. Traffic emission is the secondary contributor. It warrants further discussion.*

Response: We appreciate your comments. We discussed the PMF results in greater detail and adjusted the factor assignment to sources. The revised Figure 3 is as follows:

[Figure]

Figure 3. Contributions of identified sources to dissolved Fe, total Fe, and PM$_{2.5}$ in haze, fog, dust, and clear days by PMF model.

(Page 25)

We added discussions about industrial emissions are the largest contributor to dissolved iron in the manuscript as follows:

Figure 3 also shows that although industrial emissions (factor 5&6 or industrial emissions 1 + industrial emissions 2) contributed less than 20% to PM$_{2.5}$ in haze, fog, dust, and clear days, they were the largest contributor to dissolved Fe in haze (65.4%), fog (72.4%), dust (44.5%), and clear (62.5%) days, and they also were the largest contributor to total Fe in haze (44.2%), fog (55.0%), and clear (39.1%) days except dust days. Industrial emissions 1 (factor 5) contributed similarly to dissolved Fe regardless of weather conditions (38.9% to 43.6%, except for dusty days), while it only contributed 11.6% to 13.9% to total Fe (except dusty days). Heavy oil combustion related aerosols have the highest Fe solubility (up to 78%) from all major

Fe aerosol sources (Schroth et al., 2009; Ito et al., 2021). This may explain the much larger contribution of industrial emissions 1 to dissolved Fe than total Fe. As far as we know, there is no published data on Fe solubility in particulate matter from metal industrial emissions. Considering the dominance of iron and steel plants in total Fe emissions (Rathod et al., 2020) and the low Fe solubility in smelter ash from a steel plant (Li et al., 2017), it is difficult to understand why industrial emissions 2 (factor 6) contributes so much to dissolved Fe. Furthermore, PMF results indicated that secondary sources were the largest contributor to $PM_{2.5}$ in haze (66.2%), fog (72.3%), and clear (31.2%) days except dust days. However, the contribution of secondary sources to dissolved Fe was relatively low: 16.1% in haze days, 16.5% in fog days, 3.1% in dust days, and 5.4% in clear days.

The likely reason for the high contribution of industrial emissions 2 and the relatively low contribution of secondary sources to dissolved Fe is that PMF is unable to completely separate secondary sources of dissolved Fe (i.e., dissolved from insoluble Fe due to atmospheric processing) from primary sources. This means that some of dissolved Fe due to atmospheric processing may still be assigned to its primary factors if there is a strong co-variation between dissolved Fe and primary tracers. This suggests that the contribution of secondary sources to dissolved Fe is likely higher than that indicated by the PMF. It should also be noted that industrial emissions are outside the city and thus particles from these sources undergo long-range transport before reaching the sampling site. This provides more time for chemical processing in the atmosphere, leading to Fe solubilisation.

(Page 11, Line 325-347)

We also added discussions about the contribution of traffic emissions to dissolved Fe as follows:

As shown in Figure 3, traffic emissions contributed 10.6%, 5.8%, 18.9%, and 13.8% to dissolved Fe, and 12.7%, 7.4%, 8.1%, and 17.9% to total Fe in haze, fog, dust, and clear days, respectively. Although Fe solubility is as high as 51% in diesel exhaust and 75% in gasoline exhaust (Oakes et al., 2012), total Fe content from engine

exhaust particles is extremely low. It is more than likely that Fe from the traffic emission is associated with non-exhaust particles, which should have relatively low Fe solubility. Since traffic emissions are urban sources, which are closer to the sampling site, there are less time for them to be chemically processed in the atmosphere. This may explain why their contribution to dissolved Fe is relatively low. (Page 11, Line 318-323)

*15. Lines 302-306: As shown in Figure 5, the size of S-Fe particles in haze samples is larger than that of fog samples. The relative humidity during fog is higher than haze. Thus particles during fog are more wet, and softer, so the projected area after impacting copper is generally larger than those during haze. How are the data interpreted?*

Response: It is true that wet particles could have a larger projected area. Here the size of S-Fe particles in Figure 6 represents the dry state of individual particles on the substrate. We recognize that it is not easy to accurately measure the size of wet aerosol particles that are impacted on the substrate. The main purpose of this figure is to show acidic secondary aerosol species (e.g., sulfates and nitrate) increase the size of Fe particles. We did not intend to compare the size of S-Fe particles in different weather conditions. Therefore, we analyzed the dry state of individual particles using transmission electron microscope (TEM). We have added a description about the dry state of individual particles in the caption of Figure 6. As shown in comment 17, in order to illustrate the importance of S-Fe in Fe-containing particles in haze, fog, dust, and clear days, size distributions of Fe-rich and S-Fe particles in dust and clear were added.

[Figure]

Figure 6. Size distributions of Fe-rich (blue line) and S-Fe (green line) particles under haze (a), fog (b), dust (c), and clear (d) days. Size of S-Fe particles represents the dry state of individual particles on the substrate. The distribution pattern is normalized.

*16. As can be seen in Figure 6, the relative humidity of several fog samples is ~75%. So, the average relative humidity during the sampling period is less than 80% and visibility is less than 10 km. According to the definition in Table S2, these samples should be haze samples.*

Response: We have changed the definition of fog day in Table S2 in supplementary document. As shown in revised Table S2, in fog days, the visibility is less than 1 km, and the relative humidity is higher than 90%. We have reclassified the data according to the following criteria:

haze day: the relative humidity is less than 80%, and horizontal visibility drops below 10 km;

fog day: the relative humidity is higher than 90%, and horizontal visibility is less than 1 km.

The revised Figure 5 is as follows:

[Figure]

Figure 5. Correlations between Fe solubility and aerosol acidity/total Fe under different RH.

*17. 3.2: The authors use the data of TEM/EDS to interpret the secondary source, and regard S-Fe as an indicator of acid dissolution. The number contribution of S-Fe particles to Fe-containing particles are 76.3% and 87.1% in haze and fog days. It is not surprising. How about the clear and rain days? I speculate that the proportion of S-Fe particles have a similar contribution.*

Response: Yes, the proportion of S-Fe particles in Fe-containing particles has a similar contribution in clear days with haze/fog days. The number contribution of S-Fe particles to Fe-containing particles is 81.8% in clear days. We have added corresponding descriptions in the manuscript. To avoid misunderstanding, the number contribution of S-Fe particles to Fe-containing particles in dust days is also added in the manuscript. In rain days, we didn't collect individual particle samples, so didn't discuss TEM results in rain days.

We have changed corresponding sentences as follows:

To further support this result, a total of 688, 404, 580, and 311 individual particles in haze, fog, dust, and clear days were analyzed by TEM/EDS, respectively. In rain days, individual particle samples were not collected.

(Page 12, Line 355-356)

We further calculated the number contribution of S-Fe particles to Fe-containing particles: 76.3% in haze days, 87.1% in fog days, 78.3% in dust days, and 81.8% in clear days.

(Page 12, Line 365-366)

*18. Line 312: Under fog condition, RH ranges from 71% to 99%. Again, what is the difference in RH between fog and haze samples?*

Response: In fog days, RH is higher than 90%. In haze days, RH is less than 80%. We have revised the definition of fog day in Table S2 as follows:

The meteorological definition of fog is tiny water droplets suspended in the air, and horizontal visibility is less than 1 km, the relative humidity is higher than 90%.

*19. Line 320: The authors use SO42- and NO3- to indicate aerosol acidification. Why does NH4+ not be considered in aerosol acidification?*

Response: Yes, $NH_4^+$ can neutralize acids, so we have re-calculated the aerosol acidity. Several applications have been reported the performance of a thermodynamic equilibrium model (E-AIM model-II) (http://www.aim.env.uea.ac.uk/aim/aim.php) is accurate in evaluating the acidity nature of aerosol particles (Ansari and Pandis, 1999; Pathak et al., 2003; Yao et al., 2006; Zhou et al., 2012; Tao et al., 2019). So E-AIM model-II was selected to calculate aerosol acidity in this study. Here, aerosol acidity

was represented by in situ acidity. Because in situ acidity is a more accurate indicator of aerosol acidic nature (Pathak et al., 2009). The input data of E-AIM model-II include temperature, relative humidity, and the concentrations of $NH_4^+$, $SO_4^{2-}$, $NO_3^-$, and $H^+$. It was assumed that the concentration of $H^+ \approx 2 \times [SO_4^{2-}] + [NO_3^-] - [NH_4^+]$. We have added corresponding descriptions about E-AIM model-II in the manuscript as follows:

2.7 Aerosol acidity and liquid water content

A thermodynamic equilibrium model (E-AIM model-II) (Clegg et al., 1998) was used to calculate aerosol acidity (in situ acidity) and liquid water content, available at http://www.aim.env.uea.ac.uk/aim/aim.php. The input data include temperature, relative humidity, and the concentrations of $NH_4^+$, $SO_4^{2-}$, $NO_3^-$, and $H^+$. It was assumed that the concentration of $H^+ \approx 2 \times [SO_4^{2-}] + [NO_3^-] - [NH_4^+]$.

(Page 6, Line 161-165)

We have changed corresponding sentences as follows:

To further investigate the impact of aerosol acidification on Fe solubility, the correlation of aerosol acidity/total Fe with Fe solubility was calculated. Aerosol acidity was estimated by E-AIM model. As shown in Figure 5, aerosol acidity/total Fe and Fe solubility all show a good correlation in fog (r = 0.85, p < 0.01), haze (r = 0.56, p < 0.01), dust (r = 0.51, p < 0.05), and clear (r = 0.53, p < 0.01) days. These results further supported the above argument that the solubilisation of Fe aerosols by acids.

(Page 12, Line 367-371)

[Figure]

Figure 5. Correlations between Fe solubility and aerosol acidity/total Fe under different RH.

(Page 26)

Figure 7. Correlations between Fe solubility and liquid water content in haze (a), fog (b), dust (c), and clear (d) days.

(Page 27)

We added descriptions about the correlation analyses between dissolved Fe and liquid water content in the manuscript as follows:

Furthermore, E-AIM model was also employed to estimate liquid water content. Lower correlation between Fe solubility and liquid water content in haze (r = 0.74, p < 0.01) and clear (r = 0.65, p < 0.01) days than that in fog days (r = 0.79, p < 0.01) further supported these results (Fig. 7).

(Page 13, Line 394-396)

---

## Author Response (AR2)

**Responses to the editor comments on**

**"Sources and processes of iron aerosols in a megacity of Eastern China" by Zhu et al.**

The authors would like to thank you for the good suggestions and for giving us the chance to further improve our manuscript! We have carefully considered your comments and revised the manuscript accordingly. Below, we provide responses to the comments in blue, with changes made in the manuscript highlighted in red.

**Responses to Editor comments to the authors:**

*Thank you for the revised manuscript. We think the paper is suitable for publication if you can address this one minor comment regarding the statement that there are no data in the literature on the solubility of iron associated with PM emitted by metallurgy (lines 242-243). Recently, Mulholland et al. (2021) have reported Fe solubilities for industrial ash from a Fe–Mn alloy metallurgical plant (laboratory dissolution experiments in synthetic cloud water). These authors obtained solubilities reaching 2.5-3.0% at pH 2, after a time contact of 60 minutes (ambient temperature and UV irradiation). It might be relevant to cite these values.*

*Reference cited: In-cloud processing as a possible source of isotopically light iron from anthropogenic aerosols: new insights from a laboratory study.*

*D.S. Mulholland, Flament, P., de Jong, J., Mattielli, N., Deboudt, K., Dhont, G. and Bychkov, E. Atmospheric Environment, 2021, 259, 118505 (doi: 10.1016/j.atmosenv.2021.118505)*

Response: We have deleted lines 242-245 and added corresponding sentences as follows:

Rathod et al. (2020) suggested that metal smelting is a dominant source of anthropogenic Fe emissions. There is limited data on Fe solubility in particles from metal smelting measured in high purity water (as we did in this paper), but Mulholland et al. (2021) showed that Fe solubility of industrial ash from an Fe-Mn

alloy metallurgical plant is only about 2.8% after 60 minutes at pH = 2 synthetic solutions, suggesting a very low Fe solubility in the particles. Thus, it is unlikely that primary emissions of dissolved Fe from industrial emissions 2 (factor 6) can explain its large contribution to dissolved Fe.

(Page 8, Line 237-242)

We also added corresponding reference in the manuscript as follows:

Mulholland, D. S., Flament, P., de Jong, J., Mattielli, N., Deboudt, K., Dhont, G., and Bychkov, E.: In-cloud processing as a possible source of isotopically light iron from anthropogenic aerosols: New insights from a laboratory study, Atmos. Environ., 259, 118505, https://doi.org/10.1016/j.atmosenv.2021.118505, 2021.

(Page 14, Line 429-431)